# Highly efficient production of transgenic rats with long DNA insertions using *piggyBac* transposase mRNA and piezo-assisted microinjection

**Kohtaro Morita**[1,2]*, **Shunya Ihashi**[1], **Eiichi Okamura**[3], **Kazuya Goto**[4], **Toru Yoshihara**[1], **Arata Honda**[2,5], **Masatsugu Ema**[3], **Masahide Asano**[1]

**1** Institute of Laboratory Animals, Graduate School of Medicine, Kyoto University, Kyoto, Japan, **2** RIKEN BioResource Research Center, Tsukuba, Ibaraki, Japan, **3** Department of Stem Cells and Human Disease Models, Research Center for Animal Life Science, Shiga University of Medical Science, Otsu, Shiga, Japan, **4** Department of Regulation of Neurocognitive Disorders, Graduate School of Medicine, Kyoto University, Kyoto, Japan, **5** Center for Development of Advanced Medical Technology, Jichi Medical University, Shimotsuke-shi, Tochigi, Japan

* kohtaro.morita@riken.jp

## Abstract

In the conventional method of producing transgenic (Tg) animals, donor DNA is microinjected into the pronuclei of zygotes using a sharp glass needle. However, this approach is generally inefficient as it requires highly skilled microinjection techniques to ensure zygote survival and the transgene is incorporated into only a small proportion of the offspring. In contrast, methods based on *piggyBac* transposase (PBase) enables more efficient insertion of DNA into the genome and generation of Tg animals. The use of *piggyBac* transposase have also been examined in rats→ However, this method has not yet been fully optimized or properly characterized. In this study, we examined the microinjection of PBase mRNA and donor plasmid DNA into the pronuclei of rat zygotes using piezo-assisted microinjection. This approach resulted in high survival rates and enabled the efficient generation of Tg rats, even with long donor DNA. When the zygotes were microinjected using Piezo, over 70% were viable, and after embryo transfer, over 80% of the pups carried the transgene. Furthermore, we confirmed germline transmission to the F1 and F2 generations. We also attempted to generate a rat model of Alzheimer's using this method→ However, the protein was not detected despite mRNA expression, and the phenotype was not observed in behavioral tests. Although the generation of Alzheimer's disease model remains a challenge, our findings show that *piggyBac* transposase mRNA combined with piezo-assisted microinjection represents a simple and efficient method for producing Tg rats, even with long donor DNA.

**Data availability statement:** All data underlying the findings of this study are included in the paper and Supporting information files.

**Funding:** JSPS KAKENHI, Grant Numbers JP21H02388 (to M.E.), JP18H04883 (to A.H.), and the Kyoto University Foundation (to K.M.). The funders had no role in study design, data collection and analysis, decision to publish, or preparation of the manuscript.

**Competing interests:** The authors have declared that no competing interests exist.

## Introduction

Rats are valuable laboratory animals that are widely used in various medical studies. Rats play a significant role in neuroscience research, with many Tg strains expressing neuron-specific reporter proteins or Cre recombinases. These Tg rats are available through resources, such as the National BioResource Project-Rat (NBRP-Rat) in Japan (https://www.anim.med.kyoto-u.ac.jp/nbr/default.aspx) and the Rat Resource and Research Center (RRRC) in the United States (https://www.rrrc.us/). To generate Tg animals, conventional methods involve the microinjection of DNA fragments into the pronuclei of zygotes using a sharp glass needle. As this approach requires a high level of technical skill, the survival rate of rat zygotes microinjected by operators tends to vary widely (39.2%–39.4% [1]; 31.5%–65% [2]; 97.0%–97.5% [3]; 86.8%[4] 57.4%–75.6% [5]; 68%–88% [6]). Even when the survival rate of zygotes is high, the birth rate of the rats is not particularly high (16.6%–23.0% [1]; 3.7%–32.3% [2]; 25.1%–30% [3]; 16.7%–38.7% [5]; 3%–28% [6]), and the proportion of Tg rats among the offspring remains low (6.3%–15.2% [1]; 4.3%–33% [2]; 12.1%–13.6% [3]; 18.5% [4] 5.7%–16.7% [5]; 0%–41% [6]; 0.7%–5% [7]). The *piggyBac* transposon, which utilizes a transposase, enables the efficient random insertion of DNA sequences located between inverted terminal repeats (ITRs) into the host genome, and codon optimization of the transposase for mammals is expected to improve the efficiency of Tg animal production [8]. In addition, *piggyBac* transposase allows the insertion of multiple transgenes and facilitates the overexpression of the introduced genes, making it a suitable tool for applications requiring high levels of transgene expression. Although Tg rats have been produced using *piggyBac* transposase, the proportion of Tg offspring among live births was reported to be relatively low (14.6%–26.7%) [9]. In that study, a donor plasmid and a PBase expression vector were co-injected into the pronuclei, but the delay in PBase protein expression may have contributed to the low efficiency. Attempts to enhance PBase protein expression by microinjecting PBase mRNA together with the donor plasmid into pronuclei have yielded highly variable results, ranging from 0% to 100% efficiency, even when using the same donor construct [10]. However, as only a small number of offspring were obtained in these studies, the reproducibility of the technique, along with data on birth rates and transgene copy number, remains unclear. Recently, we successfully generated Tg mice with long DNA insertions with high efficiency using PBase mRNA and investigated the copy number of transgenes using digital PCR [11]. In addition, it has been demonstrated that piezo-assisted microinjection enables efficient nuclear membrane penetration and leads to higher embryo survival rates compared to conventional techniques in mice [12,13]. Piezo-assisted microinjection facilitates pronuclear injection with minimal damage to the zygotes, resulting in high viability.

In this study, we evaluated whether piezo-driven pronuclear co-microinjection of PBase mRNA and donor plasmid DNA is an efficient and reproducible method for generating Tg rats.

## Materials and methods

### Animals

Wistar strain rats (Crlj:WI) were purchased from Jackson Laboratory Japan (Ibaraki, Japan) and were used for embryo collection and for preparing recipient females for

embryo transfer. W-Tg(CAG-EGFP)3Ys (NBRP Rat No. 0470) rats were supplied by the NBRP-Rat, Kyoto University (Kyoto, Japan) and used as positive controls for EGFP detection in the Western blot. Female rats were euthanized by cervical dislocation following anesthesia with a mixture of three agents (0.375 mg/kg medetomidine, 2.0 mg/kg midazolam, and 2.5 mg/kg butorphanol; 0.5 mL/100 g body weight), and male rats were euthanized using carbon dioxide inhalation, in both cases to minimize suffering. All the rats were maintained in an environmentally controlled barrier room with a 12-h light cycle (7:00 AM to 7:00 PM) and 12-h dark cycle at a temperature of $23 \pm 2$ °C and humidity of $50 \pm 10\%$. This study was performed in strict accordance with the Fundamental Guidelines for Proper Conduct of Animal Experiments and Related Activities in Academic Research Institutions under the jurisdiction of the Ministry of Education, Culture, Sports, Science, and Technology of Japan and the Guidelines of Kyoto University for the Care and Use of Laboratory Animals. The protocol was approved by the Animal Experimentation Committee and Recombinant DNA Experiment Safety Committee of Kyoto University (Med Kyo 22625, 23016, and 240401).

## Preparation of donor plasmid DNA and mPBase mRNA

pUC57 mini-Myh6-NLS-EGFP-pA-(CAG-NLS-tdTomato-pA) (14.2 kb) was prepared by modifying a plasmid used in a previous study [11], and pPB-mThy1-hAPP695-mP2A-hPS1-mP2A-EGFP pA (12.9 kb) was synthesized and purchased from VectorBuilder Japan (Kanagawa, Japan). These plasmids were used as the donor DNA. The pcDNA3.1-EGFP-poly (A) 83 vector [14] containing the mammalian codon-optimized *piggyBac* transposase (mPBase) cDNA sequence [11] was used as the template plasmid for *in vitro* transcription of the mRNA. After linearization by Xho I (Takara; 1094AH, Kyoto, Japan) digestion and purification of this template plasmid containing mPBase cDNA sequence, mRNA was synthesized using a MEGAscript T7 Transcription Kit (Thermo Fisher Scientific, Massachusetts, USA; AM1333) and Cap Analog (m7G(5') ppp(5')G) (Thermo Fisher Scientific; AM8048).

## Superovulation and collection of zygotes

Superovulation was performed as in previous reports [15,16]. The adult female rats (8–15 weeks old) were injected intraperitoneally with 0.04 mg [des-Gly10, D-Ala6]-LH-RH ethylamide acetate salt hydrate dissolved in 200 µL saline (Sigma-Aldrich, St Louis, MO, US) (day 0, approximately 11:00 AM) and, with pregnant mare serum gonadotropin (PMSG; 150 IU/kg; Aska Animal Health, Tokyo, Japan) and anti-inhibin serum (100 µL; Central Research, Tokyo, Japan) 48–50 h later (day 2, 11:00 AM to 1:00 PM), followed by human chorionic gonadotropin (hCG; 75 IU/kg; Aska Pharmaceutical, Tokyo, Japan) 48 h later (day 4, 11:00 AM to 1:00 PM). The hCG-injected female and male rats were then mated in a cage. The next day, pronuclear-stage embryos were collected from mated female rats (day 5, 9:00 AM to 1:00 PM). The cells were cultured in HTF medium prior to microinjection.

## Microinjection of mPBase mRNA and donor plasmid into pronuclei of zygotes

Mixture of mPBase mRNA (final concentration 50 ng/µL) and donor plasmid DNA (final concentration 5 ng/µL) was microinjected into pronuclei of zygotes in EmbryoMax Advanced KSOM Embryo medium (Merck, Darmstadt, Germany) using an inverted microscope (IX73, Olympus, Tokyo, Japan) equipped with PMM4, a Piezo impact drive unit (Prime Tech Ltd., Ibaraki, Japan) and a micromanipulator (Narishige, Tokyo, Japan). Microinjections were performed starting between 3:00 PM and 6:00 PM. Following injection, the zygotes were placed at RT for 5 min and cultured in Rat KSOM ( ~ 50 embryos/50 µL) at 37°C under 5% $CO_2$ in air.

## Embryo transfer

Embryo transfer was performed as described in previous reports [17]. After the zygotes were cultured for 18–20 h in Rat KSOM, 10–15 of the resulting 2-cell stage embryos were transferred into the oviducts of pseudopregnant recipient

females treated with a mixture of the three anesthetic agents described above (0.5 mL/100 g body weight). The same procedure was performed with the another oviduct. Following the embryo transfer, the recipient female was administered 3% antisedan (Zenoaq, Fukushima, Japan) at the same volume as the anesthetic agent, and the rats were kept warm in a cage on a 37°C hot plate for 3 h. The number of offspring born naturally or via caesarean section was counted after 22 days of gestation.

## Detection of fluorescent protein

EGFP and tdTomato fluorescence in E10.5 embryos was visualized using a stereo microscope (SMZ800N, Nikon, Japan) equipped with a camera (CHUNICHI SUWA Optoelectronics; SS500-MC, Nagoya, Japan) and a fluorescence adapter (OLFAS-2 RB-GO and GR; BEX, Tokyo, Japan). The tdTomato fluorescent protein in newborn pups and adult rats was visualized using a handy light (OptoCode, Tokyo, Japan).

## Genotyping from genomic DNA

Genomic DNA was extracted by phenol/chloroform extraction and ethanol precipitation after digestion of the fetus or tail tissue in a buffer solution containing Proteinase K and sodium dodecyl sulfate (SDS) at 37°C overnight. Genotyping was performed through conventional PCR and electrophoresis. Amplifications were run in a Thermal Cycler Dice TP600 (Takara) and a T100 Thermal Cycler (Bio-Rad, Hercules, CA, USA). The reaction parameters were as follows: *tdTomato*, one cycle at 94°C for 1 min and 35 cycles at 98°C for 10 s, 58.7°C for 15 s, and 68°C for 30 s using Tks Gflex DNA Polymerase; *Egfp*, one cycle at 94°C for 1 min and 35 cycles at 98°C for 10 s, 68°C for 30 s using Tks Gflex DNA Polymerase; *5'ITR-Thy1 promoter* and *rGB pA-3'ITR*, one cycle at 95°C for 10 min and 35 cycles at 98°C for 30 s, 55°C for 1 min, and 72°C for 30 s using Bio TAQ (BIO-21040, NIPPON Genetics, Tokyo, Japan). The PCR products were electrophoresed on 2% agarose gels with TBE and visualized using Midori Green Advanced (1:10,000; NE-MG04; NIPPON Genetics) or Gel Red Nucleic Acid Gel Stain (1:10,000; 41003; Biotium, CA, USA).

## Assessment of transgene copy number

Droplet digital PCR (ddPCR) was performed using the QX200 Droplet Digital PCR System (Bio-Rad). A mixture of 5' FAM, 3' Iowa Black FQ (IBFQ)-labeled quencher in combination with the internal ZEN quencher and GFP sequence primer was synthesized and purchased from IDT. A quencher and primers were used to detect the GFP sequences (S1 Table). Mixtures of 5' HEX, 3' IBFQ-labeled quencher and Rat Rpp30 sequence primer and 5' 6-FAM, 3' IBFQ-labeled quencher and human APP (hAPP) sequence primer were synthesized and purchased from Bio-Rad (S1 Table). The genomic DNA, primer, probe, and Hae III restriction enzyme concentration were 30 ng/µL, 900 nM, and 250 nM, 0.5 U/µL respectively, for a total reaction volume of 20 µL per sample. Droplets were formed using a QX200 droplet generator (Bio-Rad). The conditions maintained in C1000 Touch Thermal Cycler (Bio Rad) were 95°C for 10 min; 94°C for 30 s and 53°C for EGFP and Rpp30 or 56°C for hAPP and Rpp30 for 1 min, 40 cycles; 98°C for 10 min; and 4°C at infinite hold. The droplets were analyzed using a QX200 Droplet Reader (Bio-Rad).

## RT-PCR

Total RNA was isolated from the brain, liver, heart, spleen, lung, and kidney of the rats by using the RNeasy Plus Mini Kit (50) (QIAGEN; 74134), and cDNA was synthesized using the SuperScript IV One-Step RT-PCR System (Thermo Fisher Scientific, 12595025). *Actb* was used as the internal control. The reaction mixture without the RT enzyme was used as a negative control. Following RT, the prepared cDNA samples were amplified and analyzed through PCR. Amplification was performed using a T100 Thermal Cycler (Bio-Rad). The reaction parameters were as follows: one cycle at 50°C for 10 min, one cycle at 98°C for 2 min and 35 cycles at 98°C for 10 s, 68°C for *EGFP,* 58.7°C for *tdTomato* and 57.9°C for

*Actb* for 10 s, and 72°C for 15 s, and one cycle at 72°C for 5 min. The RT-PCR primers used are listed in S1 Table. The RT-PCR products were electrophoresed in the same manner as the genomic DNA.

## Western blotting

Samples from the cerebral cortex were subjected to SDS-polyacrylamide gel electrophoresis. A 3.4-mg sample of cerebral cortex was added to 500 µL of 1 x Laemmli sample buffer (Bio-Rad) containing 5% 2-mercaptoethanol (Nacalai tesque, Kyoto, Japan). The sample was homogenized and denatured by heat shock for 5 min at 95°C, and each sample was electrophoresed by SDS-PAGE on 12% TGX FastCast Gel (1610175; Bio-Rad) at 100 V for 1 h, and then transferred onto a 0.2 µm PVDF membrane (1704156, Bio-Rad) using Trans-Blot Turbo transfer system (Bio-Rad). The membranes were blocked using EzBlock Chemi (Atto, Tokyo, Japan) at RT for 1 h, and washed two times with 0.05% PBS-T for 10 min, then incubated at 4°C overnight with rabbit anti-GFP antibody (1:10,000; ab183734; Abcam, UK) and goat anti-β-Actin antibody (1:5,000; sc-1616; Santa Cruz, TX, USA) in 0.05% PBS-T containing 10% blocking buffer, respectively. The membranes were washed twice with 0.05% PBS-T for 10 min, incubated with donkey anti-rabbit IgG-horseradish peroxidase (HRP) conjugate (1:100,000; AP182P; Millipore, Burlington, MA, USA) and donkey anti-goat IgG HRP conjugate (1:100,000; AP180P; Millipore) at RT for 1 h, washed twice with PBS-T for 10 min, and developed using ECL Prime Western Blotting detection reagent (Cytiva, Tokyo, Japan). Blot images were acquired using a C-DiGit Blot Scanner (LI-COR, NE, USA) and Image Studio software.

## Immunohistochemistry

Endogenous peroxidase activity in the brain sections was quenched by incubating them in methanol containing 0.3% hydrogen peroxide for 5 minutes. After rinsing in PBS, nonspecific binding was blocked with 3% bovine serum albumin (BSA) for 30 minutes at room temperature. The sections were then incubated overnight at 4 °C with a monoclonal anti–Aβ antibody (12B2; IBL, Maebashi, Japan). The following day, the sections were processed using the Histofine Mouse Stain Kit MAX-PO (M) (Nichirei Biosciences, Tokyo, Japan) according to the manufacturer's protocol. Immunoreactivity was visualized with 3,3'-diaminobenzidine (DAB), and the sections were counterstained with hematoxylin.

## Fear conditioning test

The fear conditioning test was conducted over three days using 13-month-old 5xFAD Tg (n = 5) and 11-weeks-old wild-type (n = 12) rats. The test comprised three training and testing phases. On the first day, conditioning was conducted in a box (W, 327 mm; D, 250 mm; H, 284 mm; 100 lx) in which the rats were allowed to move freely for 8 min. In the conditioning phase, white noise (55 dB) and electrical foot shock (0.3 mA, 2 s) were presented simultaneously three times: 120–150 s, 240–270 s, and 360–390 s. On the second day, freezing performance was measured for 5 min in the same box, without white noise or foot shock, to assess contextual fear memory. On the third day, rats were placed in a different triangular box (W, 350 mm; D, 350 mm; H, 390 mm; 10 lx) to measure cued fear memory. The freezing performance was monitored for 6 min, and the same white noise used in the conditioning phase was presented from 180 to 210 s. All equipment and analysis software were purchased from O'HARA Co., Ltd., Tokyo, Japan.

## Statistical analysis

We performed the Mann-Whitney *U* test with an α level of 0.05 to determine possible statistically significant differences.

The detailed protocol for generating transgenic rats has been deposited in protocols.io (https://doi.org/10.17504/protocols.io.dm6gpmnwdgzp/v1).

## Results

### Generation of Tg rats by piezo-assisted co-injection of mPBase mRNA and donor plasmid DNA

We examined whether Tg rats could be efficiently generated by co-injecting mPBase mRNA and donor plasmid DNA into the pronuclei of zygotes using a piezo-assisted microinjection method (Fig 1A). First, we prepared a donor plasmid encoding cardiac-specific *Egfp* and ubiquitously expressed *tdTomato*, both of which have been shown to function in mice (Fig 1B) [11]. Microinjection of donor plasmid DNA and mPBase mRNA mixtures into the pronuclei of zygotes resulted in a survival rate of 72.8% (Table 1).

The rate of litters obtained from embryo transfer was 22.5%, and 26.1% of the births were stillbirths. Moreover, not all of the surviving pups produced from Myh6-EGFP- and CAG-tdTomato-injected embryos reached weaning age (Table 1). Genotyping by PCR revealed that 15 of the 18 (83.3%) samples examined were Tg-positive. When embryos confirmed to have developed to the fetal stage were included, 23 of 26 (88.4%) were Tg-positive (Table 1, Figs 1C and S1). These results suggest that mPBase mRNA and large donor DNA injected into the pronuclei of zygotes by the piezo-assisted method were efficiently inserted into the genome with high efficiency. Next, to examine the copy number of the donor DNA insertion in Tg rats using mPBase, we performed droplet digital PCR (ddPCR) and determined the copy number in these Tg rats. Although some rats had low copy numbers, many had high copy numbers (Fig 1D, S2 Table). These results suggest that our method enables the efficient production of high-copy number Tg rats, even with large DNA.

### Expression of transgene in F0 Tg rats

To determine whether the transgene is expressed and functioned in the Tg rats, we confirmed mRNA expression and fluorescence in various organs. *Egfp* mRNA was detected only in the heart of the five organs examined, and *tdTomato* was detected in all organs (Figs 2A and S2). Consistent with our previous report in mice, EGFP fluorescence was observed only in the heart, and tdTomato fluorescence was detected throughout the body in E10.5 fetuses (Fig 2B) [11]. At the time of caesarean section, some offspring were already stillborn, and the weaning rate was also slightly low. Furthermore, 60% of the weaned offspring died starting at 6 weeks post-weaning and the number of surviving offspring decreased to 40% by 12 weeks (Fig 2C). In addition, surviving rats had copy numbers ranging from 0.19 to 1.11, whereas deceased animals, including stillbirths, had significantly higher copy numbers (Fig 2D). These findings suggest that death is caused by excessive amounts of fluorescent proteins. Based on the above results, we successfully inserted the transgene into the rat genome with high efficiency and expressed the protein.

### Germline transmission of the transgene from F0 rats

To confirm whether the transgene is inherited by the next generation, Myh6-EGFP and CAG-tdTomato F0 males and females that survived to sexual maturity were crossed with the wild type (WT), resulting in seven of the 15 offspring being *Egfp*-positive by genomic PCR (Fig 3A). However, one of the seven offspring was *tdTomato*-negative (No. 9), and tdTomato fluorescence was observed in all six rats, except for the one identified as EGFP-positive and tdTomato-negative by PCR analysis (Figs 3 B, C and S3). As only F0 rats with approximately one copy of *Egfp* survived, the copy number in Tg-positive F1 rats ranged from 0.8 to 1.2 (Fig 3D, S3 Table). In F2 embryos, EGFP expression was specifically observed in the heart, and tdTomato was expressed throughout the body; EGFP- and tdTomato-positive fetuses were also confirmed to be transgene-positive by genomic PCR (Figs 3E, F and S3). Furthermore, tdTomato fluorescence was detected after birth (Fig 3G). These findings indicate that functional transgenes are expressed in subsequent generations.

### Attempt to generate disease model rats

Because rats have good memories and are commonly used in behavioral experiments, we attempted to generate a rat model of Alzheimer's disease. The Tg rats carried a 9.9-kb sequence encoding a gene with five mutations found in human

**A**

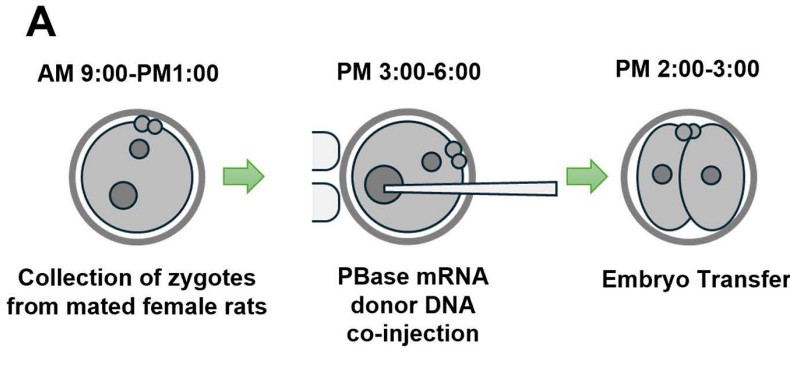

AM 9:00-PM1:00

Collection of zygotes
from mated female rats

PM 3:00-6:00

PBase mRNA
donor DNA
co-injection

PM 2:00-3:00

Embryo Transfer

**B**

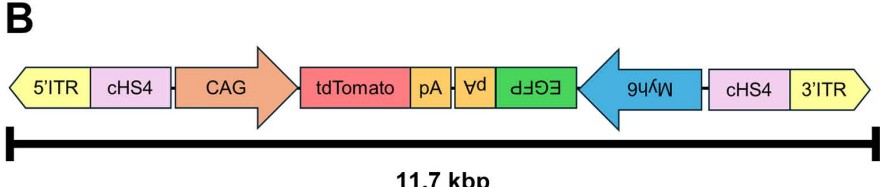

11.7 kbp

**C**

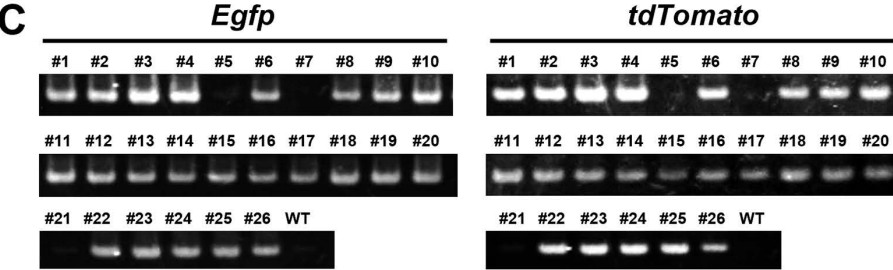

**D**

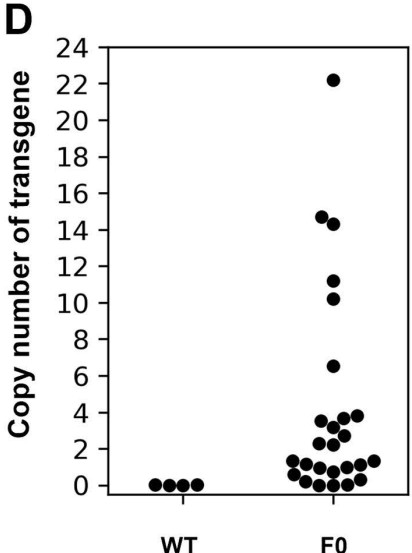

**Fig 1. Generation of Myh6-EGFP, CAG-tdTomato Tg rats by mPBase mRNA and donor plasmid DNA pronuclear microinjection.** (A) Schematic diagram of the method to introduce mPBase mRNA and donor plasmid DNA. (B) Schematic representation of Myh6-EGFP, CAG-tdTomato donor plasmid DNA. Only the region between two ITRs is described. (C) Genotyping of Myh6-EGFP, CAG-tdTomato Tg rats. (D) Copy number analysis (*Egfp*) of Tg harboring Myh6-EGFP, CAG-tdTomato transgene.

**Table 1. Production efficiency of Myh6-EGFP, CAG-tdTomato Tg rats.**

| Types of injected DNA | No. (%) of surviving injected-zygotes | No. (%) of 2-cell embryos | No. of 2-cell embryos transferred | No. (%) of pregnant females | No. (%) of offspring in pregnant females | No. (%) of live offspring at birth | No. (%) of stillborn offspring at birth | No. (%) of weaned offspring | No. (%) of Tg offspring |
|---|---|---|---|---|---|---|---|---|---|
| Myh6-EGFP, CAG-tdTomato | 142/195 (72.8±1.80) | 102/142 (71.8±0.49) | 102 | 4/4 (100) | 23/102 (22.5±6.92) | 17/23 (73.9±8.48) | 6/23 (26.1±8.48) | 12/23 (52.2±9.85) | 15/18* (83.3±12.5) |

*Total no. of 6 stillborn offspring and 12 weaned offspring.

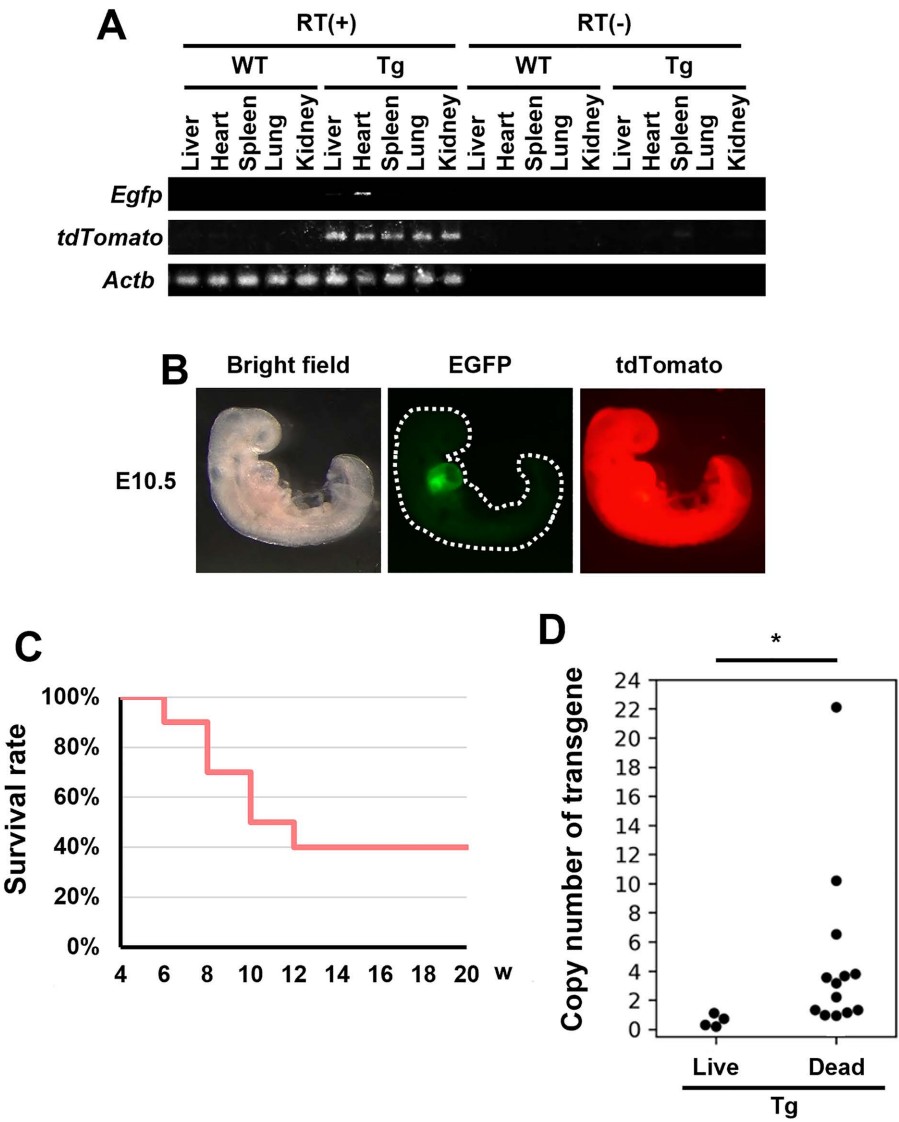

**Fig 2. Phenotype of Myh6-EGFP, CAG-tdTomato Tg rats.** (A) mRNA expression of *Egfp*, *tdTomato*, and *Actb* in five organs. (B) Fluorescent images of E10.5 embryos harboring Myh6-EGFP, CAG-tdTomato transgene. Bright field (left), EGFP fluorescence (middle), tdTomato fluorescence (right) images are shown. EGFP and tdTomato positive (6/6 embryos). (C) Survival rate of Myh6-EGFP, CAG-tdTomato Tg rats after weaning. (D) Copy number analysis (*Egfp*) of live and dead Tg rats harboring Myh6-EGFP, CAG-tdTomato transgene at 12-weeks-old. *$p < 0.05$.

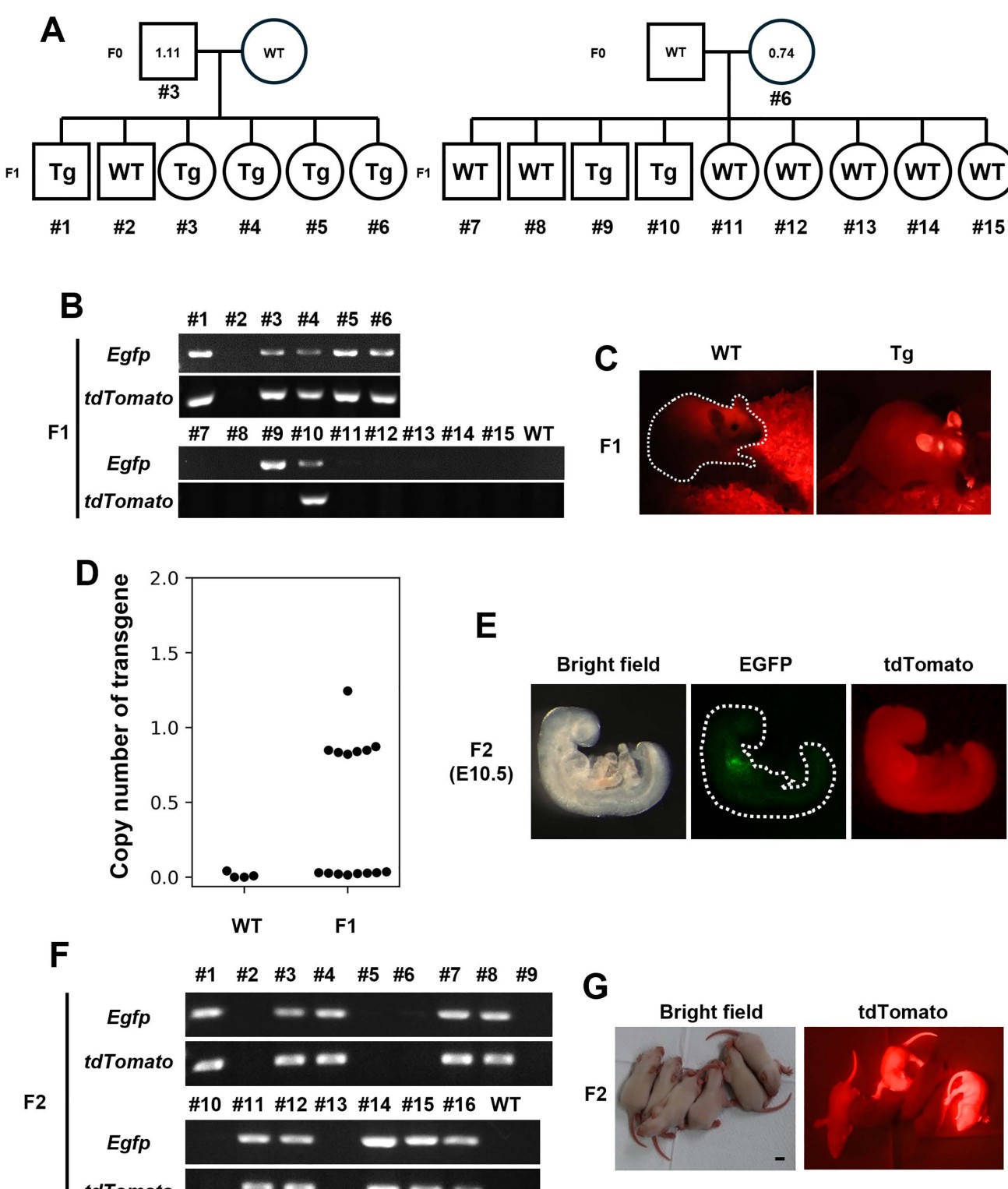

**Fig 3. Confirming germline transmission of the transgene.** (A) A mating test was performed to confirm germline transmission of the transgene. Two of the surviving Tg rats were crossed with the wild type (WT); the pedigree of line #3 and #6 are shown. Squares indicate males, and circles indicate females. (B) Genotyping of F1. Individuals with amplified transgenes were determined to be heterozygous and the results were shown in (A). (C)

Fluorescent images of tdTomato in adult F1 rat. tdTomato positive (6/7 Tg rats). (D) Copy number of the transgene was determined by ddPCR using DNA obtained from the tail tip. (E and F) Fluorescent images and genotyping of E10.5 F2 Tg embryos harboring Myh6-EGFP, CAG-tdTomato transgene. Bright field (left), EGFP fluorescence (middle), tdTomato fluorescence (right) images are shown. EGFP and tdTomato positive (10/10 Tg embryos). (G) Fluorescent images of tdTomato in F2 pups. Scale bar = 1 cm.

familial Alzheimer's disease (5xFAD), which include a mutant human amyloid beta (Aβ) precursor protein (APP) cDNA sequence (K670N/M671L (Swedish) + I716V (Florida) + V717I (London), and a mutant human presenilin 1 (PSEN1) cDNA sequence ( M146L + L286V). Tg mice with these five mutations (5xFAD) have been created and are expected to serve as a model for early onset Alzheimer's disease [18,19]. Here, we attempted to generate 5xFAD Tg rats and examined the effect of transgene expression on learning behavior (Fig 4A). We achieved a 96% Tg-positive rate, with an average copy number of 13.3 (range: 0.69–58.2) (Figs 4B, C and S4, Tables 2 and S4). In addition, when we compared the copy number of F1 between the Tg rat with the highest copy number (58 copies) and the WT, the copy number of F1 was approximately half that of F0 (Fig 4D, S5 Table).

In the F0 analysis, the transgene-derived mRNA was also detected in the cerebral cortex (S5A Fig). However, despite using rats with more than 10 copies of the transgene, EGFP protein and accumulated Aβ was not detected in cerebral cortex at 9-month-old 5xFAD Tg rats (S5B and C Fig). Furthermore, in the fear conditioning test, no significant differences were observed between 13-month-old 5xFAD Tg rats and the younger 11-week-old WT rats (S6 Fig). Although distinct phenotypes were not observed in the 5xFAD Tg rats generated in this study owing to the lack of protein expression from the transgenes, we demonstrated that it is possible to produce Tg rats carrying up to 58 copies of the transgene.

## Discussion

In previous reports, Tg rats were produced with 33%–100% efficiency by using DNA with plasmid sizes of 7.4 and 5.9 kb, together with *piggyBac* transposase mRNA, although the size of the transgene was not specified [10]. The authors of that study reported that only a few offspring were obtained, and the efficiency of Tg rat production was unclear [10]. Although we could not obtain Tg rats with 100% efficiency, we succeeded in introducing approximately 10 kb of large-size donor DNA (Plasmid DNA size: 12.9 kb to 14.2 kb) to the rat genome simply and efficiently by combining the *piggyBac* and piezo-assisted microinjection (83.3%–96.4%) (Tables 1 and 2). In a previous report, Tg production efficiency was found to be high in rats but low in mice (0%–31%) [10]. Because the pronuclear stage in rats is longer than that in mice [20,21], mPBase mRNA has sufficient time to be translated, which may explain why Tg rats can be efficiently generated by simply co-injecting donor DNA and mPBase mRNA. To efficiently produce Tg mice, mPBase mRNA is introduced into zygotes by electroporation, allowing sufficient time for translation into proteins before the injection of donor DNA plasmids into the pronuclei of the mouse zygotes [11]. Furthermore, given that 71.5%–72.8% of zygotes survived after piezo-assisted micro-injection with good penetration efficiency (Tables 1 and 2), it may be possible to stably obtain injected oocytes without advanced micromanipulation skills. Therefore, piezo-assisted microinjection is expected to become an important tool for generating Tg and KI animals.

Previously, we demonstrated that IVF oocytes can be efficiently used to produce genome-edited rats [16]. However, when IVF is performed in the morning to obtain fertilized oocytes with pronuclei, pronuclear formation occurs at night, requiring microinjection into the pronucleus during late working hours [17]. In contrast, when superovulated females are naturally mated, fertilized oocytes with pronuclei can be collected between morning and noon (approximately 9:00 AM to 12:00 PM) the following day, allowing microinjection to be performed in the early evening (approximately 4:00 PM to 6:00 PM). Therefore, naturally mated zygotes are more suitable for Tg rat production because they involve fewer procedural steps and yield a higher rate of embryonic development. Thus, the procedure for generating Tg animals may be simpler and involve fewer steps in rats than in mice.

 

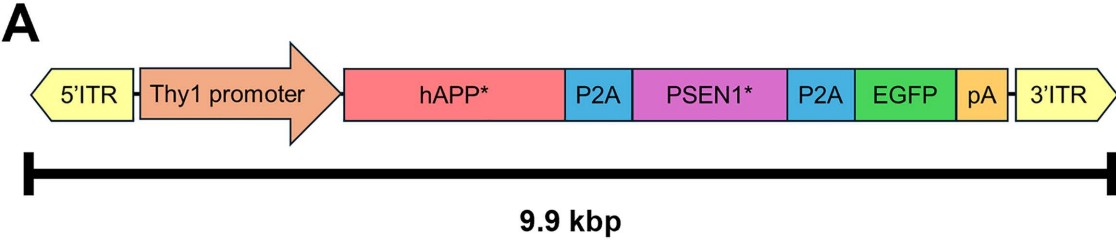

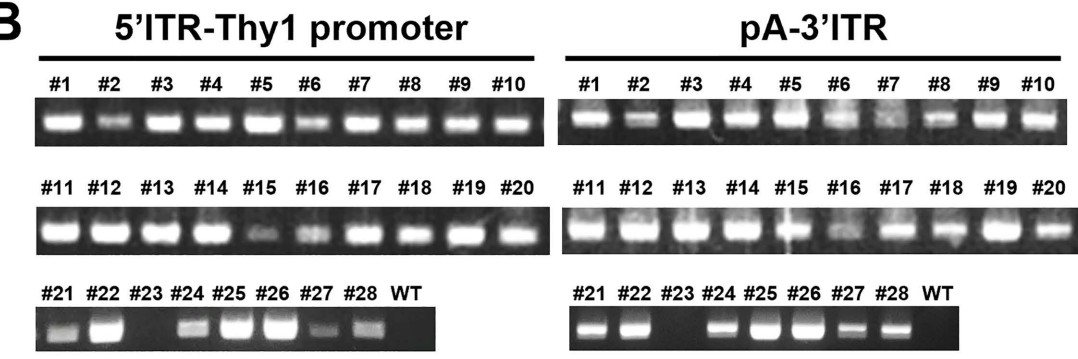

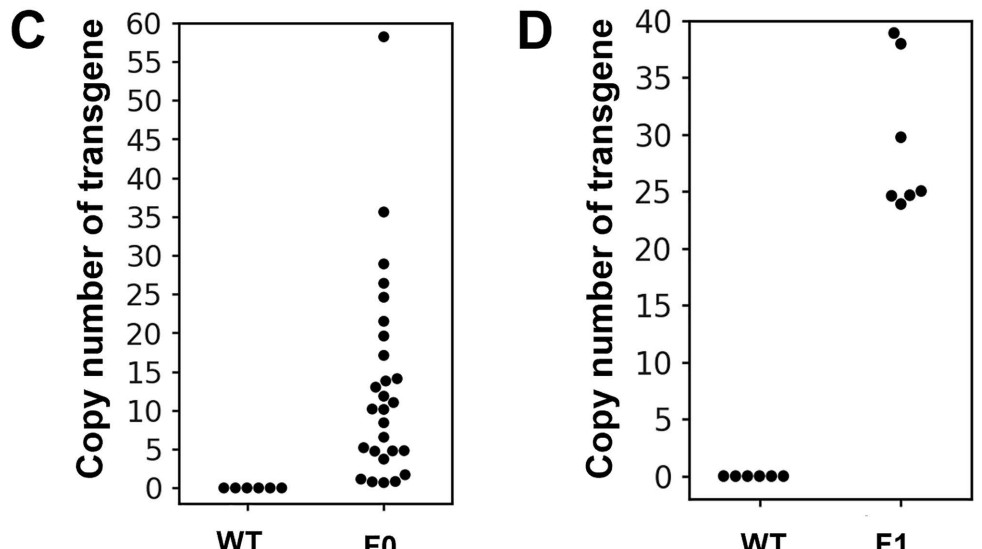

**Fig 4. Generation of 5xFAD Tg rats.** (A) Schematic representation of Thy1-hAPP, PSEN1, EGFP donor plasmid DNA. Only the region between two ITRs is described. (B) Genotyping of 5xFAD Tg rats. (C and D) Copy number analysis of Tg harboring Thy1-hAPP, PSEN1, EGFP transgene (hAPP) in F0 and F1.

**Table 2. Production efficiency of 5xFAD Tg rats.**

| Types of injected DNA | No. (%) of surviving injected-zygotes | No. (%) of 2-cell embryos | No. of 2-cell embryos transferred | No. (%) of pregnant females | No. (%) of offspring in pregnant females | No. (%) of live offspring at birth | No. (%) of still-born offspring at birth | No. (%) of weaned offspring | No. (%) of Tg offspring |
|---|---|---|---|---|---|---|---|---|---|
| Thy1-hAPP, PSEN1, EGFP | 133/186 (71.5) | 113/133 (85.0) | 80 | 4/4 (100) | 28/80 (35) | 28/28 (100) | 0/28 (0) | 28/28 (100) | 27/28 (96.4) |

As expected, Myh6-EGFP, CAG-tdTomato Tg rats expressed EGFP specifically in the heart and tdTomato ubiquitously throughout the body (Fig 2A and B). However, some fetuses died during embryogenesis and postnatal growth. In addition, high copy numbers were not obtained in surviving Tg rats (Fig 2D). Although these fluorescent proteins have NLS for localization in the nucleus, the overexpression of fluorescent proteins with transport signals is known to interfere with intracellular transport systems [22]. In addition, a previous study shows that mice with high tdTomato expression under the CAG promoter die after weaning, with a survival rate of only 0%–40% up to 12-weeks-old, similar to our results [23] (Fig 2C). Therefore, it is possible that some rats died because the overexpressed fluorescent proteins accumulated in the nucleus and interfered with certain functions. The transmission of the transgene to the next generation was confirmed (Fig 3). However, among the offspring derived from F0 with a copy number of 0.74, WT animals were predominant, whereas heterozygotes were underrepresented (Fig 3A). Although further investigation is required, this bias may be attributed to the lower transgene copy number (0.74) compared to F0 Tg rats with a copy number of 1.11.

In the previous study, 5xFAD Tg mice show no significant differences in the fear conditioning test, which is used to evaluate Alzheimer's disease [18]. Furthermore, several studies have shown that 5xFAD Tg mice exhibit different phenotypes, making them difficult to study in humans [19]. Although we aimed to generate 5xFAD rats, we did not obtain a phenotype in 5xFAD Tg rats (S5 and S6 Figs). Previously studied 5xFAD Tg mice are double Tg, with hAPP and PSEN1 transgenes introduced separately [24]. In contrast, we designed donor plasmid DNA to link hAPP, PSEN1, and *Egfp* via P2A sequences to ensure equal expression levels. Therefore, the resulting protein may have undergone abnormal processing and degradation, as a result of which Aβ was not generated. Another possibility is that the transgene includes the 5' untranslated region (UTR) of human, which may not be efficiently recognized by the translational machinery in rats, unlike in mice [24,25].

With the revision of the plasmid constructs, the methodology we have presented here is expected to contribute to the efficient generation of Tg rats as models for Alzheimer's and other diseases. In conclusion, our *piggyBac* transposase mRNA and piezo-assisted microinjection technique provides a simple and efficient platform for generating Tg rats with broad potential for producing disease models and other complex genetic modifications, although it still requires piezo equipment and basic skill in micromanipulation, and embryonic toxicity may occur depending on the transgene copy number.

## Supporting information

**S1 Fig. Genotyping of Myh6-EGFP, CAG-tdTomato Tg rats in F0.** Images of uncropped and minimally adjusted agarose gels corresponding to Fig 1C.
(TIF)

**S2 Fig. RT-PCR for *Egfp*, *tdTomato*, and *Actb* in five organs.** Images of uncropped and minimally adjusted agarose gels corresponding to Fig 2A.
(TIF)

**S3 Fig. Genotyping of Myh6-EGFP, CAG-tdTomato Tg rats in F1 and F2.** Images of uncropped and minimally adjusted agarose gels corresponding to Fig 3B and F.
(TIF)

**S4 Fig. Genotyping of 5xFAD Tg rats in F0.** Images of uncropped and minimally adjusted agarose gels corresponding to Fig 4B.
(TIF)

**S5 Fig. Expression of the transgene in 5xFAD Tg rats.** (A) Analysis of transgene mRNA expression in cerebral cortex by RT-PCR. Images of uncropped and minimally adjusted agarose gels. (B) Analysis of transgene protein expression in cerebral cortex by western blot. PC: positive control (CAG-EGFP Tg rat). Images of uncropped and minimally adjusted blots. (C) Immunohistochemistry of Aβ in cerebral cortex. Scale bar: 1 mm.
(TIF)

**S6 Fig. Fear conditioning test.** 5xFAD Tg rats at 13-month-old and WT rats at 11-week-old were used. *$p < 0.05$, **$p < 0.01$, *n.s.*: not significant.
(TIF)

**S1 Table. Primers for RT-PCR and primers/quencher for ddPCR used in this study.**
(XLSX)

**S2 Table. Copy number of Myh6-EGFP, CAG-tdTomato Tg rats (F0).**
(XLSX)

**S3 Table. Copy number of Myh6-EGFP, CAG-tdTomato Tg rats (F1).**
(XLSX)

**S4 Table. Copy number of 5xFAD Tg rats (F0).**
(XLSX)

**S5 Table. Copy number of 5xFAD Tg rats (F1).**
(XLSX)

## Acknowledgments

We thank the members of the Institute of Laboratory Animals, Graduate School of Medicine, Kyoto University, Mr. Yoshio Sasaoka, Mr. Kento Morita, and Ms. Saki Wajima, for their assistance with the experiments and rat breeding. We also thank Dr. Kimiko Inoue and Dr. Atsuo Ogura of RIKEN BRC for providing the opportunity to conduct experiments at their institute and the National BioResource Project-Rat for providing the rat strain (W-Tg(CAG-EGFP)3Ys). Fear conditioning test was performed at the Medical Research Support Center, Graduate School of Medicine, Kyoto University. We also thank Editage (www.editage.jp) for English language editing.

## Author contributions

**Conceptualization:** Kohtaro Morita, Eiichi Okamura, Toru Yoshihara, Arata Honda, Masatsugu Ema.

**Formal analysis:** Kohtaro Morita.

**Funding acquisition:** Kohtaro Morita, Arata Honda, Masatsugu Ema, Masahide Asano.

**Investigation:** Kohtaro Morita, Shunya Ihashi, Kazuya Goto.

**Methodology:** Kohtaro Morita.

**Project administration:** Kohtaro Morita.

**Resources:** Eiichi Okamura, Masatsugu Ema, Masahide Asano.

**Supervision:** Arata Honda, Masatsugu Ema, Masahide Asano.

**Validation:** Kohtaro Morita.

**Visualization:** Kohtaro Morita.

**Writing – original draft:** Kohtaro Morita.

**Writing – review & editing:** Kohtaro Morita, Shunya Ihashi, Eiichi Okamura, Kazuya Goto, Toru Yoshihara, Arata Honda, Masatsugu Ema, Masahide Asano.

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
