## [Decision Letter · Decision Letter 0]

7 Nov 2025

Dear Dr. Morita,

Thank you for submitting your manuscript to PLOS ONE. After careful consideration, we feel that it has merit but does not fully meet PLOS ONE’s publication criteria as it currently stands. Therefore, we invite you to submit a revised version of the manuscript that addresses the points raised during the review process.

We look forward to receiving your revised manuscript.

Kind regards,

Wilfried A. Kues, Ph.D.

Academic Editor

PLOS ONE

Journal Requirements:

3. To comply with PLOS ONE submissions requirements, in your Methods section, please provide additional information regarding the experiments involving animals and ensure you have included details on (1) methods of sacrifice, and (2) efforts to alleviate suffering.

“JSPS KAKENHI, Grant Numbers JP21H02388 (to M.E.), JP18H04883 (A.H.), and the Kyoto University Foundation (to K.M.)

“The authors wish to acknowledge members of the Institute of Laboratory Animals, Graduate 324 School of Medicine, Kyoto University, Mr. Yoshio Sasaoka, Mr. Kento Morita, and Ms. Saki 325 Wajima, for their assistance with the experiments and rat breeding. We also thank Dr. Kimiko 326 Inoue and Dr. Atsuo Ogura of RIKEN BRC for providing the opportunity to conduct experiments 327 at their institute and the National BioResource Project-Rat for providing the rat strains (W328 Tg(CAG-EGFP)3Ys). Fear conditioning test was performed at the Medical Research Support 329 Center, Graduate School of Medicine, Kyoto University. This study was supported in part by JSPS 330 KAKENHI, Grant Numbers JP21H02388 (to M.E.), JP18H04883 (A.H.), and the Kyoto 331 University Foundation (to K.M.). We would like to thank Editage (www.editage.jp) for English 332 language editing.”

“JSPS KAKENHI, Grant Numbers JP21H02388 (to M.E.), JP18H04883 (A.H.), and the Kyoto University Foundation (to K.M.)”

Reviewer's Responses to Questions

**Comments to the Author**

1. Is the manuscript technically sound, and do the data support the conclusions?

Reviewer #1: Yes

2. Has the statistical analysis been performed appropriately and rigorously?

Reviewer #1: Yes

3. Have the authors made all data underlying the findings in their manuscript fully available?

Reviewer #1: Yes

4. Is the manuscript presented in an intelligible fashion and written in standard English?

Reviewer #1: Yes

Reviewer #1: Review Manuscript PONE-D-25-52370:

Highly efficient production of transgenic rats with long DNA insertions using piggyBac transposase mRNA and piezo-assisted injection

Overall Assessment

This manuscript presents a study describing a highly efficient method for generating transgenic rats using piggyBac transposase mRNA and donor plasmid DNA combined with piezo-assisted pronuclear microinjection. The authors demonstrate high zygote survival rates in comination with a high transgenic efficiency, even when using large donor DNA constructs (>10 kb). The method is clearly described and experimentally validated across F1 and F2 generations.

The study is technically sound, addresses a real bottleneck in rat transgenesis. Although it should not be understated that the technique still requires a high level of technical skill in microinjection and familiarity with piezo-assisted systems. While the method may lower the barrier for experienced embryologists by improving reproducibility and zygote survival, it does not entirely eliminate the need for precise micromanipulation expertise.

2. Validity of the Methods

• The experimental design is clear and reproducible, with detailed descriptions of plasmid preparation, microinjection, embryo transfer, and genotyping.

• Ethical approval and animal welfare considerations are properly stated.

• The use of droplet digital PCR (ddPCR) for copy-number quantification is appropriate and provides solid validation.

In the Materials and Methods section, please add which strain was used for embryodonation and which one for the embryo transfer etc. (Line 70). In addition, include the anesthesia protocol including pain management, as the cited source is not publicly accessible (line 113).

3. Soundness of the Results and Data Interpretation

• The high integration rate and germline transmission are convincing and strongly support the authors’ claims.

• The observed correlation between high copy number and postnatal lethality is interesting and consistent with known effects of overexpression from strong promoters.

• The 5xFAD rat model section is less successful: despite mRNA expression, the lack of protein and phenotype suggests translational inefficiency or construct design issues (e.g., promoter incompatibility or 5′-UTR sequence effects). This section could be condensed or moved to the Supplementary Information, as it distracts from the main technical findings.

4. Appropriateness of the Conclusions

The conclusions are well supported by the data. The authors correctly state that the piggyBac mRNA plus piezo-injection method allows for efficient production of transgenic rats with long DNA inserts.

However statements about the simplicity and broad applicability of the method should be tempered with acknowledgement of limitations (e.g., requirement for piezo equipment, variable copy numbers and possible embryonic toxicity).

5. Clarity and Presentation

The manuscript is generally well written and logically structured.

6. Recommendation

Decision: With minor revisions, this manuscript is suitable for publication in PLOS ONE.

**Do you want your identity to be public for this peer review?** For information about this choice, including consent withdrawal, please see our Privacy Policy

Reviewer #1: No

---

## [Author Response · Author response to Decision Letter 1]

21 Nov 2025

We have attached a separate file containing our point-by-point responses to the reviewer comments.

---

## [Decision Letter · Decision Letter 1]

7 Dec 2025

Highly efficient production of transgenic rats with long DNA insertions using piggyBac transposase mRNA and piezo-assisted microinjection

PONE-D-25-52370R1

Dear Dr. Morita,

We’re pleased to inform you that your manuscript has been judged scientifically suitable for publication and will be formally accepted for publication once it meets all outstanding technical requirements.

Kind regards,

Wilfried A. Kues, Ph.D.

Academic Editor

PLOS One

Additional Editor Comments (optional):

Reviewers' comments:

Reviewer's Responses to Questions

**Comments to the Author**

Reviewer #1: All comments have been addressed

2. Is the manuscript technically sound, and do the data support the conclusions?

Reviewer #1: Yes

3. Has the statistical analysis been performed appropriately and rigorously?

Reviewer #1: Yes

4. Have the authors made all data underlying the findings in their manuscript fully available?

Reviewer #1: Yes

5. Is the manuscript presented in an intelligible fashion and written in standard English?

Reviewer #1: Yes

Reviewer #1: (No Response)

**Do you want your identity to be public for this peer review?** For information about this choice, including consent withdrawal, please see our Privacy Policy

Reviewer #1: **Yes:** Wiebke Garrels

---

## [Editor Report · Acceptance letter]

PONE-D-25-52370R1

PLOS One

Dear Dr. Morita,

I'm pleased to inform you that your manuscript has been deemed suitable for publication in PLOS One. Congratulations! Your manuscript is now being handed over to our production team.

Kind regards,

on behalf of

Dr. Wilfried A. Kues

Academic Editor

PLOS One